# Determinants of Buying Produce on Short-Video Platforms: The Impact of Social Network and Resource Endowment—Evidence from China

**Shu Jiang, Zhanpeng Wang, Zilai Sun \*** and **Junhu Ruan**

College of Economics and Management, Northwest A&F University, Yangling 712100, China
*   Correspondence: drszl@nwafu.edu.cn; Tel.: +86-1334-2289-901

**Abstract:** In the wake of the COVID-19 pandemic, selling by short video has become a new online selling model that enhances the communication between buyers and sellers. Therefore, it is necessary to identify the key factors influencing consumers' purchase of agricultural products on short-video platforms. Additionally, it is also important to figure out the influencing mechanism and action path. Specifically, based on the 'Stimulus-Organism-Response (SOR)' framework and structural equation model, we delineate and empirically test hypotheses regarding the effects of key components on consumers' purchase intentions and behaviors. The key components refer to three external stimuli of consumers' social network, sellers' resource endowment, and both sides' infrastructure development levels. Simultaneously, we analyze the mediating role of consumers' perceived value and perceived risk between external stimuli and consumers' purchase intentions. This paper argues that short-video merchants improving the influence of their stores and platforms strengthening supervision and management are the keys to ensuring stable growth in consumers' willingness to purchase agricultural products sold on short videos and promoting the development of the short-video live industry.

**Keywords:** social network; resource endowment; agricultural products; consumers' purchase behavior; short videos





## 1. Introduction

The outbreak of the COVID-19 pandemic at the end of 2019 has led to severe disruptions in food production and transportation worldwide and difficulties in selling agricultural products on offline channels [1]. As a result, e-commerce platforms such as JD.com and pinduoduo.com have begun to focus on fresh and agricultural sales in the sector. At the same time, selling agricultural products on short-video platforms has become a no-brainer for many farmers. In addition, the Chinese government unveiled the "No.1 central document on 22 February 2022, namely, "Opinions of the State Council of the Central Committee of the Communist Party of China on the Key Efforts to Comprehensively Promote Rural Revitalization in 2022", outlining key tasks to comprehensively push forward rural vitalization this year. The document calls for efforts to implement the project of "digital business to promote agriculture". It also emphasizes the need to accelerate the standardized and healthy development of live-streaming e-commerce of agricultural and sideline products.

In practice, "Internet + agricultural products" has become a trendy mainstream of agricultural product sales in the past two years. According to the data statistics of Chinese authoritative websites, the transaction scale of China's live-streaming e-commerce market reached $368.9 billion in 2021, an increase of 82.87% over the previous year. Among them, the fresh e-commerce transaction scale reached $73.55 billion, up 27.92% year-on-year. In addition to this, the penetration rate of live e-commerce reached 10.15% in 2021, breaking through double digits and growing by as much as 18.02%. Over the next few years, there will still be high growth. "Live +" will become the new normal of e-commerce, short video

and the live e-commerce "people and goods" range is rapidly expanding, and the short video and live way will become the "standard" of e-commerce.

Compared with the traditional sales model, selling through short videos breaks the limitations of time and space, reduces the marketing costs of agricultural products, and improves the consumer buying experience. As the new carriers of fragmented information, short video belongs to the category of a dynamic display of online products, which can enhance the attention value of the display content and the effective perception of the value of products by consumers compared with static display [2]. Therefore, more and more short-video platforms have attracted much attention. In this context, how short-video platforms can improve their communication effect, how to increase the attractiveness of their products to consumers, and how to improve consumer satisfaction, are considerations that have become the focus of short-video research and the urgent need to solve the problem.

Short videos have the characteristics of fast dissemination and strong mass participation, and more and more consumers choose to purchase products on short-video platforms. Existing studies show that the vividness and interactivity of product presentation affect consumers' willingness to purchase [3]. The influence of product information on trust is well established in both traditional and online environments, and the more detailed the product images and descriptions, the more trust is gained, which in turn, affects consumers' purchase intentions [4]. Lavie and Tractinsky found that good visual design of information presented in an e-tailing environment can influence consumer pleasure and purchase intention by studying environmental psychology models [5]. Socio-demography and situational factors have an impact on online purchase intention [6]. Grohmann et al. pointed out that a good virtual tactile experience on a short video platform can increase consumers' emotional involvement and stimulate their purchase intentions [7]. Additionally, Yoo and Kim explored the effect of online displays on consumer responses from the perspective of mental imagery, which increase their behavioral intention to purchase by triggering a positive emotional response to the product presentation [8].

In summary, most of the existing studies analyzed the impact of marketing tools and content presentation of short videos on consumers' purchasing behavior from the perspective of short videos themselves. However, only a few scholars studied the impact of social network, resource endowment, and infrastructure on consumers' purchasing intentions on the platforms from the perspective of external factors. In addition, vegetable and fruit agricultural products are perishable, which are difficult to store and transport, and the economic benefits are also low. As a result, it is more typical to study the influence of the above-mentioned external factors on consumers' purchasing behaviors of vegetable and fruit agricultural products on short-video platforms.

In this study, we construct a structural equation model based on the analysis of the SOR(Stimulus-Organism-Response) theoretical model to explore the influence of social network, resource endowment, infrastructure and other factors on the purchase behavior of short video consumers of fruits and vegetables. Compared with the existing studies, the contributions of this paper are as follows:

(1) In terms of research content, the existing literature focus on a single discussion from short videos themselves, for example, Jiang and Benbasat analyzed consumers' purchase intentions based on product display and video presentation [3]. At the same time, most scholars, such as Lavie and Tractinsky, Grohmann et al., Yoo and Kim, explored the mechanism level of purchase behavior, revealing less about the internal influence mechanism [5,7,8]. In particular, there is less literature exploring how perceived risk affects consumers' purchases of agricultural products on short-video platforms. In this study, focusing on vegetable and fruit agricultural products, we introduce three external stimulus variables and explore the mechanisms of social network, resource endowment, and infrastructure on consumers' purchase behavior of agricultural products on short-video platforms, which enriches the research on short-video platform sales.

(2)  In terms of research methods, existing literature adopt a variety of empirical analysis methods to study the issue. Yet, to our knowledge, there has been relatively little literature combining the SOR model and structural equation method to study the factors influencing consumers' purchasing behaviors on short-video platforms. Therefore, we adopt the structural equation approach based on the SOR theoretical model to analyze the correlation between the variables. On this basis, we explore whether there is a mediating effect of perceived value and perceived risk between the three external stimulus variables and consumers' willingness to purchase. Then we discuss the factors that influence consumers' purchasing behavior of agricultural products, so as to improve the income level of farmers.

The paper begins by presenting the introduction and is followed by the literature review of this research. Then, the conceptual framework and hypotheses are discussed and the research method adopted in this paper is introduced in the next section. The final section is a discussion of the findings and policy recommendations.

## 2. Literature Review

### 2.1. Aspects of the Short-Video Platform Marketing Model

In recent years, due to the development of information technology and the epidemic, short videos have developed rapidly, and a series of short-video platforms have emerged, attracting a large amount of traffic and attention. Taking TikTok as an example, Xu et al., analyzed the reasons why short video apps became popular rapidly and the problems in marketing [9]. Gao used the 4R marketing theory (4Rs refer to Relevance, Reaction, Relationship, and Reward. The 4R marketing theory aims at building consumer loyalty and believes that as the market evolves, companies need to build new proactive relationships between them and their customers at a higher level and in a more effective way) to explore the marketing model between agricultural e-commerce and short video, and analyzed the problems of product quality and content marketing in this model, and proposed optimization strategies [10]. Öztamur and Karakadılar explored how to use short-video marketing to increase the performance of SMEs [11]. Makushkin et al. took short videos as an Internet marketing tool to attract potential college students (applicants) to use YouTube services and theoretically analyzes the modes and types of occurrence of video marketing [12]. Qian explored the driving effect brought by the model of live online video sales and showed that the current stage of the existence of the marketing model is a single and cost problem, and, finally, also put forward the idea of innovative agricultural economic development [13]. It can be seen that in recent years, the short-video marketing fever is gradually rising, especially after the epidemic offline sales channels are blocked, and short-video marketing is becoming more and more popular.

### 2.2. Aspects of Consumers' Willingness to Buy on Short-Video Platforms

Short videos are preferred by many people for their advantages of fast dissemination, low cost, and strong mass participation, and many consumers buy products on short-video platforms. Jiang and Benbasa confirmed the effect of vividness and interactivity of product display on consumers' willingness to return to the website and purchase products through a modeling study [3]. Huang and Suo examined four factors that influence Chinese live e-commerce consumers' impulse purchase decisions from a consumer perspective, including price promotions, time pressure, interpersonal interaction, and visual appeal [14]. Hao et al. noted that the rapid expansion of online transactions and the boom in Internet technology have increased the importance of effective product video displays for online retailers. Short videos can increase customers' purchasing intention by improving the information they received [15]. Saha et al. found that both consumer satisfaction and payment efficiency of online purchases enhanced their willingness to pay and contributed to an insight into the impact of cost-saving efficiency on online customer satisfaction and repurchase intention [16]. Dominici et al. explored the impact of sociodemographic and situational factors on online purchase intentions through a survey to help marketers and

retailers determine marketing strategies [6]. Roggeveen et al. indicated that dynamic display formats increase engagement with the product/service experience in a manner presumably similar to the actual product experience [2]. These essays above showed that consumers are mostly willing to make purchases on short-video platforms compared to traditional offline sales channels because the dynamic video display and the interactivity between consumers and merchants increase their ability to access information.

### 2.3. Aspects of Consumer Purchase Behavior Influencing Factors

Thanks to the rapid development of modern technology, short-form video shopping has become an increasingly accepted method, and there are many factors that influence consumers in making their choices. Liu et al. used short social media videos as the main research object to explore companies' brand attitudes towards consumers using short social media marketing [17]. Xu pointed out that the current product information display methods on e-commerce platforms are mainly graphic, short video, and live streaming, and the research direction of male consumers' purchase intentions is more likely to be influenced by live streaming [18]. Yoo and Kim explored the impact of online displays on consumer responses from the perspective of mental imagery, which increased consumers' behavioral intentions by triggering positive emotional responses to product presentations [8]. Deng et al. analyzed the impact of gender and generational differences on consumers' purchase behavior of wine on TikTok by examining their purchases [19]. Xiao et al. explored the online shopping context that determines consumers' purchase intentions and identified four cues that promote such consumption behavior in cross-border e-commerce, such as online promotion cues, content marketing cues, personalized recommendation cues, and social commentary cues. A theoretical model based on the cue-utilization theory and the stimulus–organism–response model is proposed to introduce these four cues and brand familiarity into the analysis of the impact of cross-border online shopping (CBOS) on consumers' purchase intentions [20].

It can be seen that consumers' purchase intentions on short-video platforms are influenced by numerous factors, both positive and negative. However, the discussion of the influencing factors mainly stays on the short videos themselves, including their video content and marketing methods, etc. Few studies focus on the influence of consumers' social networks and merchants' resource endowments on consumers' purchase intention from the perspective of both consumers and merchants. Therefore, in this paper, we fill this gap by focusing on the impact of external stimuli and consumers' perceived value and risk on final purchase intentions.

### 2.4. Aspects of Structural Equation Modeling

Due to the deteriorating ecological environment in the Yangtze River basin, Chen and Wang conducted a study on the willingness of consumers to purchase wild freshwater fish clearly, and they used structural equation modeling to analyze data from a survey of 1235 consumers in eight provinces of the Yangtze River basin. The results of the study showed that personal norms, attitudes, and environmental concerns were the most influential factors [21]. Rezai et al. examined the willingness of consumers to purchase functional foods so that decisions could be made from a sell-side perspective that would be more favorable to companies engaged in this industry [22]. In doing so, the authors used structural equation modeling and interviewed 2004 households through a structured questionnaire, and eventually drew their conclusions. In addition to this, Munerah et al. used the structural equation method in their study of non-green consumers' purchase intentions toward green beauty products, and based on this, the method was optimized and validated using quantitative methods of structural equation modeling with partial least squares, and concluded that the attribution of responsibility, social norms, etc., have a significant effect on personal norms [23].

In summary, it can be seen that when studying consumers' willingness to purchase a certain good, especially when there are many measured variables, many scholars choose

the structural equation model to conduct their research or use it as a basis for further research by combining other methods. Therefore, when we study the purchase intention of consumers on short-video platforms, we also learn from the practice of predecessors and adopt the structural equation modeling method to achieve the expected research purpose.

## 3. Conceptual Framework and Hypotheses

### 3.1. A Model of Consumer Buying Behavior

The SOR model is also known as the Stimulus–Organism–Response theory. In 1974, Mehrabian and Russell introduced this model based on environmental psychology to analyze the influence of the environment on human behavior [24]. The model explains that under the influence of the external environment, consumers' psychological awareness changes, resulting in convergent or avoidance behavior [25]. The SOR model has three variables, namely S: external stimuli (including external macro and micro stimuli); O: organism, and often includes the cognitive and affective variables of the consumer; R: response variable (the behavior of the consumer as a result of internal changes under the influence of external stimuli). In 1982, Donovan et al. adapted the model and applied the SOR model to shopping situations for the first time, suggesting that the store environment elicits a basic effective state of pleasure–arousal–control in individuals, which in turn influences individuals to produce profit–avoidance behavior [26]. Huang took the SOR model as a research framework, combined with social capital theory and immersion theory, to explore the influence of affective and reactive factors on the impulse buying behavior of social commerce consumers [27]. Drawing on the Stimulus–Organism–Response (SOR) framework, a model is proposed and tested around the impacts of smart service scape dimensions (aesthetics, superior functionality, social presence, perceived interactivity, and perceived personalization) on smart consumer experience co-creation [28]. In the context of e-tailing, Eroglu and Machleit have argued that stimuli are the sum of all triggers seen and heard by online consumers, and that emotions and perceptions act as mediating states between stimuli and individual responses [29].

Based on the previous review of the relevant model construction and theoretical studies of the SOR theoretical model, as well as the review of the research content related to the purchase behavior of short-video consumers, we propose the following model construction ideas.

(1)  S: social network, resource endowment, and infrastructure are selected as external stimulus variables;
(2)  O: perceived value and perceived risk are selected as mediating variables;
(3)  R: consumer purchase behavior is selected as the response variable of the research model.

Based on the above, we give a conceptual model diagram of the factors that influence consumer purchasing behavior, as shown in Figure 1.

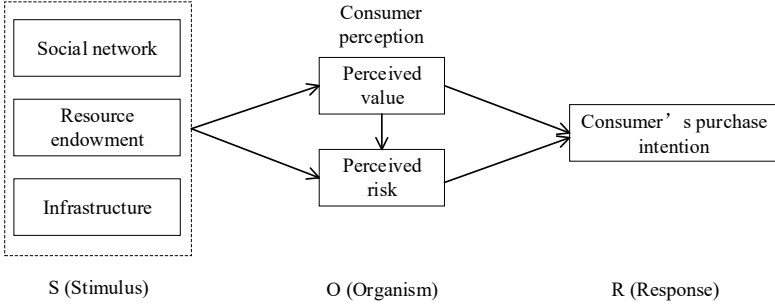

**Figure 1.** A consumer buying behavior model based on SOR.

### 3.2. Research Hypotheses

3.2.1. The Role of External Stimuli in Influencing Consumer Perception and Purchase Behavior

Social network in this paper mainly includes the number of likes and retweets of short videos, the usage of people around and the influence of opinion leaders, etc., Dellarocas empirically showed that online word of mouth is an important determinant of consumer purchases, with positive word of mouth having a positive effect on consumers and negative word of mouth having a negative effect [30]. Due to the virtual nature of online shopping, consumers' perceived product value is confirmed through information sharing with other users, especially with opinion leaders who are at the center of the structural hole for information help and corroboration [31]. During a pandemic, consumers prefer the targeting of socially close beneficiaries of marketing actions to socially distant ones [32]. On short-video platforms, high likes and retweets often lead to a herd mentality among consumers, who are more willing to make purchases. The use of people around them then increases consumers' interest in the platform and increases their willingness to purchase to some extent. Some studies showed that opinion leaders use their expertise to actively participate in online community discussions, which helps to drive community members to participate in interactions and elicit cognitive and emotional resonance from followers, gaining consumers' trust, and thus, increasing their willingness to purchase [33]. At the same time, a higher level of online word of mouth and trust is likely to reduce consumers' perception of online purchase risk to a certain extent, thus, facilitating their purchase behavior. Based on the above theoretical analysis, the following hypotheses are proposed in this paper.

**H1a:** *Social network has a significant negative effect on perceived risk.*

**H1b:** *Social network has a significant positive impact on consumer purchase behavior.*

Since in this study we focus on vegetable and fruit agricultural products, the main measures of resource endowment (size of planting area), human resource endowment (level of e-commerce training), and technical resource endowment (level of agricultural mechanization) are selected. From the farmers' perspective, the information feedback mechanism of the online trading platform creates the possibility for farmers to anticipate the next year's sales, which, in the long run, helps farmers to adjust their planting scale and planting structure, increase their willingness to plant and participate in e-commerce, and then meet the higher purchasing demand of consumers and increase their purchasing behavior [34]. In addition, compared with traditional trading methods, rural e-commerce has stronger user stickiness, and growers can improve their understanding of e-commerce through regular e-commerce training, which in turn can expand their region-specific sales and purchasing behavior to the national level. It can be inferred from this that the number and level of e-commerce training for farmers at the origin of agricultural products affects consumers' willingness to purchase, considering the consumer perspective. If the farmers in that area regularly attend e-commerce training and will have a higher quality of e-commerce sales, consumers tend to subjectively perceive the produce of that location as trustworthy, decreasing their perceived risk. Similarly, the higher the level of agricultural mechanization, the more consumers trust the technical resource endowment of the place, which in turn increases the purchasing behavior. Based on the above theoretical analysis, the following hypotheses are proposed in this paper.

**H2a:** *Good resource endowment has a significant negative effect on perceived risk.*

**H2b:** *Good resource endowment has a significant positive effect on consumer purchasing behavior.*

The infrastructure in short-video sales mainly refers to the express logistics of the place of delivery and the place of receipt of agricultural products, the distance between the places of delivery and receipt of agricultural products, and the storage conditions of cold storage. Moga explored the level of development of agricultural development and

computer technology in Romania, and then analyzed the factors influencing e-commerce for online services, and found that production factors, information technology factors, resource factors, and Internet penetration had a significant impact on the development of rural e-commerce [35]. Rohani et al. studied farmers' satisfaction with the partnership system of chicken companies and showed that the production infrastructure services provided by chicken companies significantly and positively increased farmers' loyalty to the partnership of chicken companies [36]. For vegetable and fruit agricultural products, the convenience of logistics and express delivery as well as the storage condition of cold storage has an impact on both the perceived value and the perceived risk of consumers; the better the facilities, the higher the perceived value and the lower the perceived risk, which will have a significant positive impact on consumers' purchasing behaviors. Therefore, the following hypotheses are proposed in this paper.

**H3a:** *Infrastructure improvements have a significant negative effect on perceived risk.*

**H3b:** *Infrastructure improvements have a significant positive impact on consumer purchasing behavior.*

### 3.2.2. The Mediating Role of Perceived Value between External Stimuli and Consumers' Willingness to Buy

Dodds et al. pointed out that price, brand, and store information all have an impact on consumer purchase decisions [37]. Woodruff was the first to suggest that there is a correlation between brand perception and consumer perception, and further analyzed the relationship between brand perception and consumer perception such as quality and value, etc., [38]. Parasuraman et al. argued that consumers' perceptions of products and services come from personal experience and the loss or value gained after service, and this value is the most important reason for customers' purchase intentions [39]. In the short-video platform, perceived value can build a bridge between consumers and merchants, and consumers can choose whether to make a purchase or not through the perceived value of fruits and vegetables. Therefore, we propose the following hypothesis.

**H4:** *Perceived value mediates between social network, resource endowment, infrastructure, and consumer purchase behavior.*

### 3.2.3. The Mediating Role of Perceived Risk between External Stimuli, Perceived Value, and Consumer Purchase Intentions

Agheshlouei et al. found that perceived value and perceived risk have a mediating role in the relationship between store image and consumer purchase intentions [40]. The popularity of short video platforms has allowed consumers to make purchases on short video platforms, and social network relationships can, to some extent, affect consumers' perceived risk of that product, and thus, final purchase behaviors. Similarly, resource endowment and infrastructure can also affect consumers' perceived online shopping risks at different levels, which further influences consumers' purchasing behaviors. Therefore, the following hypothesis is proposed in this paper.

**H5:** *Perceived risk mediates between social network, resource endowment, infrastructure, perceived value and consumer purchase behavior*

## 4. Methodology
### 4.1. Study Design
#### 4.1.1. Questionnaire Design

In order to ensure the reliability and validity of the variables, we refer to the measurement question items of potential variables commonly used in related literature at home and abroad. When selecting the variables and designing the questionnaire, we use them as the theoretical basis. At the same time, we supplement and modify the questionnaire to con-

sider the actual situation of consumers making purchases through short-video platforms. The specific measures and question items of each variable in the final questionnaire are shown in Table 1. The questionnaire measures are based on a five-point Likert scale, with 1 indicating "strongly disagree", 2 indicating "disagree", 3 indicating "average", 4 indicating "agree", and 5 indicating "strongly agree". After completing the questionnaire design, we invited experts and scholars in this field to review the content of the questionnaire and put forward suggestions for modification. Then, we improved and finalized the questionnaire according to the suggestions.

**Table 1.** Relevant variables and descriptions.

| Potential Variables | Measurement Problem Items | Title Source |
|---|---|---|
| Social network | SN1 I prefer to choose the video merchants with a higher number of likes and retweets for purchase<br>SN2 I feel more interested when people around me are also using short videos to make purchases<br>SN3 I am easily influenced by the more persuasive people around me when buying on short videos | Chevalier and Mayzlin [41]<br>Dellarocas [30] |
| Resource Endowment | RE1 When I buy through short videos, I consider the scale of its growing area<br>RE2 I prefer to buy agricultural products from areas where farmers have a higher quality of e-commerce training<br>RE3 I would be more willing to buy if the local agricultural mechanization level is high | Nakasone [34]<br>Mishra and Park [42] |
| Infrastructure | IF1 When I buy through short videos, I pay attention to the logistics of the place where the product is received and shipped<br>IF2 When I buy through short videos, I consider the distance from where the product is shipped<br>IF3 I would be more willing to buy if there were local storage facilities such as cold storage | Moga [35]<br>Rohani et al. [36] |
| Perceived Value | PV1 I think the fruit and vegetable produce I buy on the short-video app is worth the money I paid<br>PV2 I think shopping on the short-video platform is economical and cost effective<br>PV3 I think it is attractive to shop on short-video apps | Grohmann et al. [7]<br>Trautwein and Lindenmeier [43] |
| Perceived Risk | PR1 I would be worried about the quality of the produce purchased on the short-video app<br>PR2 I would be worried about receiving an item that does not meet my expectations<br>PR3 I would be worried about financial losses when returning products purchased on short-video apps | Sweeney et al. [44]<br>Wu et al. [45] |
| Purchase Intention | PI1 When I have a need, I am willing to buy produce directly on the short video app<br>PI2 For the same product, I prefer to buy on the short-video app<br>PI3 The short-video app has a significant impact on my purchasing behavior | Mehta et al. [46]<br>Kukar-Kinney et al. [47] |

The interpretation of the variables is as follows.

1. External stimulus variables

Social network: the theoretical basis of social network is derived from the Six Degrees of Separation theory, that is, society is a huge network composed of a variety of relationships. The significance of social network analysis is that it can provide a precise quantitative analysis of various relationships, thus, providing a quantitative tool for the construction of some middle-level theories and the testing of empirical propositions, and even building a bridge between "macro and micro". In the case of short-video platform purchases, social network relations are mainly reflected in the number of likes and retweets of short videos, the usage of surrounding people and the influence of opinion leaders. When consumers are browsing short videos, they are often influenced by the number of people watching

and liking a short video, and thus, choose whether to stay on that page to watch and make a purchase. In addition, opinion leaders often have a greater impact on consumers' purchasing behaviors.

Resource endowment: resource endowment, also called factor endowment, is proposed by Swedish economists Brown Köchel and Olin on the basis of Ricardian comparative advantage theory, mainly used to explain the reasons why countries have comparative cost advantages in producing goods involved in international trade exchange, specifically refers to the abundance of a country or a region with various factors of production such as labor, capital, land, management, technology, etc. Since this paper mainly explores the sales of vegetable and fruit agricultural products on the short-video platform, the size of the cultivation area, the level of farmers' e-commerce training, and the level of agricultural mechanization will be selected as the main independent variables to be measured. In this way, we can better understand consumers' attention to the resource endowment of agricultural products' production areas when they choose whether to buy fruits and vegetables on short-video platforms.

Infrastructure: Infrastructure refers to the material engineering facilities that provide public services for social production and residents' lives, and is a public service system used to ensure the normal conduct of social and economic activities in a country or region. In terms of short-video platform sales, infrastructure mainly refers to the express logistic situation at the place of delivery of agricultural products, as well as the place of receipt, the distance from the place of delivery of agricultural products, and the storage conditions of cold storage. In the study, Zhu et al. pointed out that the improvement of logistic factors and transportation conditions had a significant positive impact on the adoption of different marketing channels by small farmers [48]. In the study on potato cultivation, Akello et al. noted that incorrect storage methods, as well as storage techniques, can lead to low yields and low quality of potatoes [49]. When consumers buy fruits and vegetables, people tend to pay more attention to the transportation and storage situation of express delivery because of their characteristics, such as not being easy to preserve and being perishable. Therefore, infrastructure is of great importance to study the behavior of short-video consumers in purchasing vegetable and fruit agricultural products.

2.  Mediating variables

Perceived value: perceived value generally includes the perceived functional price, perceived emotional value, and perceived social value. Perceived value plays a partial mediating role in the relationship between unfamiliar word of mouth and purchase intention, which can further enhance consumers' emotional attitudes toward the product. In the purchase behavior of short-video platforms, consumers' perceived values largely affect their purchase intention and purchase behavior. Therefore, this paper selects perceived value as a mediating variable to study consumers' purchasing behavior.

Perceived risk: perceived risk usually includes perceived time risk, functional risk, physical risk, financial risk, social risk, and psychological risk. When consumers make purchases of fruits and vegetables on the platform, they mostly consider whether the produce will be damaged during transportation, the freshness of the produce, and whether the quality of the produce is guaranteed, which means that consumers receive a series of perceived risks when they choose whether to purchase or not. Therefore, this paper will consider that perceived risk may have the mediating effect of rational thinking between external stimuli and generating purchase intention.

3.  Response variables

Consumer purchase behavior is selected as the response variable in the model. Since this paper wants to explore the behavior of consumers purchasing vegetable and fruit agricultural products on short-video platforms, consumer purchasing behavior is selected as the outcome variable in the research model.

### 4.1.2. Data Collection

The data collection adopts a combination of online and offline methods. The online questionnaire is released through a professional questionnaire survey website and posted in places where short-video consumers gather a lot; the offline research takes the form of random sampling. The offline research site was selected as a representative agricultural high-tech industry demonstration area—the Yangling Demonstration Zone. In the Yangling Demonstration Zone, 5 representative communities were selected for field research and data sampling, 30 questionnaires were distributed in each community, and a total of 150 questionnaires were collected from 5 communities. From there, the actual local situation was projected according to the respective sampling intensity according to the random sampling principle. Finally, 269 questionnaires were collected online and 150 questionnaires were collected offline, totaling 419, and the ratio of online and offline questionnaires collected was 1.79; excluding illogical and invalid questionnaires, there were 406 questionnaires with 96.9% validity.

### 4.1.3. Descriptive Statistics of the Sample

The basic situation of 406 valid samples is shown in Table 2: the survey sample of women accounted for a relatively large proportion, 61.58%, and relatively few men; the majority of interviewees were between the ages of 13 and 62, with the largest proportion being in the 42–62 age group; the education level of the respondents is mainly high school and undergraduate, accounting for 69.70% of the survey sample, followed by junior high school, accounting for 21.67% of the survey sample, elementary school and below, and masters and above accounted for 7.14% and 1.48%, respectively; the monthly income level, concentrated distribution in 2000–5000 yuan (including 5000 yuan), accounting for 35.96% of the survey sample, the rest of the monthly income level of the sample number is relatively small; more than 90% of the survey sample are aware of live e-commerce and short-video apps downloaded on the phone, indicating that most people have heard of live e-commerce, and download short-video apps to watch and browse.

**Table 2.** Descriptive statistical analysis of the sample.

| Statistical Indicators | Classification Indicators | Number of Samples | Proportion/% |
|---|---|---|---|
| Gender | Male | 156 | 38.42 |
| | Female | 250 | 61.58 |
| Age | $13 \leq a \leq 26$ | 134 | 33.00 |
| | $27 \leq a \leq 41$ | 122 | 30.05 |
| | $42 \leq a \leq 62$ | 144 | 35.47 |
| | $63 \leq a$ | 6 | 1.48 |
| Academic qualifications | Elementary School and below | 29 | 7.14 |
| | Junior High School | 88 | 21.67 |
| | High School | 126 | 31.03 |
| | Undergraduate | 157 | 38.67 |
| | Masters and above | 6 | 1.48 |
| Monthly income | $0 \leq i \leq 1000$ yuan | 73 | 17.98 |
| | $1000 < i \leq 2000$ yuan | 88 | 21.67 |
| | $2000 < i \leq 5000$ yuan | 146 | 35.96 |
| | $5000 < i \leq 10{,}000$ yuan | 74 | 18.23 |
| | $10{,}000$ yuan $< i$ | 25 | 6.16 |
| Knowing live e-commerce | Yes | 385 | 94.83 |
| | No | 21 | 5.17 |
| Downloading short-video apps | Yes | 369 | 90.89 |
| | No | 37 | 9.11 |

### 4.2. Method

Structural equation modeling (SEM) is a collection of statistical techniques. The model is mainly used to study the relationship between one or more independent variables (continuous or discrete) and one or more dependent variables (continuous or discrete) [50]. In regression analysis or path analysis, even though the graphs of statistical results show multiple dependent variables, the regression coefficients or path coefficients are calculated for each dependent variable one by one. The graphs appear to consider multiple dependent variables at the same time, but the presence of other dependent variables and their effects are ignored when calculating the effect or relationship on a particular dependent variable.

In this study, there are three external stimulus variables: social network, resource endowment, and infrastructure, two mediating variables are consumers' perceived values and perceived risks, and one response variable is consumers' purchase intentions. Therefore, this is a multiple-to-multiple study and the simple regression model cannot be used, so we resort to structural equation modeling for a more comprehensive study.

In addition, if we want to know the correlation degree between latent variables, each of which is measured by multiple indicators or topics, a common practice is to use factor analysis for each latent variable to first calculate the relationship between the latent variable (i.e., factor) and the topic (i.e., factor loading), which in turn yields a factor score as the observed value of the latent variable, and then the factor score is used as the correlation coefficient between the latent variables. These are two independent steps. In the structural equation, these two steps are performed, simultaneously, i.e., the factor–topic relationship and the factor–factor relationship are considered, simultaneously. Therefore, the model fits well with the study of this paper.

### 4.2.1. Test of Reliability and Validity

Before conducting exploratory factor analysis (EFA), we calculate KMO values and conduct Bartlett's sphere test. The results show a KMO value of 0.900 and a probability of significance of 0.000 for the statistical value of Bartlett's sphere test, indicating that the data are suitable for exploratory factor analysis [51]. The sample data are sampled for six factors by the criterion of eigenvalues greater than one, explaining 76.506% of the variance. The validation factor analysis (CFA) is used to test the reliability and validity of the variables, and the results are shown in Table 3. The Cronbach's alpha coefficient and CR values of each factor in the scale are higher than 0.700, indicating that the reliability of the scale is good [51]. The average extracted variance (AVE) of each factor is higher than 0.500, indicating that the scale has a good convergent validity [52].

**Table 3.** Factor standard loadings, AVE, CR, and alpha values.

| Factor | Indicators | Standard Load | AVE | CR | $\alpha$ |
|---|---|---|---|---|---|
| Social network | SN1<br>SN2<br>SN3 | 0.791<br>0.811<br>0.706 | 0.594 | 0.814 | 0.789 |
| Resource Endowment | RE1<br>RE2<br>RE3 | 0.798<br>0.744<br>0.853 | 0.639 | 0.841 | 0.842 |
| Infrastructure | IF1<br>IF2<br>IF3 | 0.599<br>0.776<br>0.810 | 0.539 | 0.775 | 0.796 |
| Perceived Value | PV1<br>PV2<br>PV3 | 0.813<br>0.840<br>0.843 | 0.692 | 0.871 | 0.897 |
| Perceived Risk | PR1<br>PR2<br>PR3 | 0.798<br>0.793<br>0.767 | 0.618 | 0.829 | 0.825 |
| Purchase Intention | PI1<br>PI2<br>PI3 | 0.814<br>0.786<br>0.728 | 0.604 | 0.820 | 0.832 |

In order to test the discriminant validity of the model, we use the Fornell–Larcker criterion to evaluate the discriminant validity by comparing the square root of the AVE values of each factor and the factor correlation coefficients. The logic of the Fornell–Larcker method is based on the idea that a construct shares more variance with its associated indicators than with any other construct [52]. The results are shown in Table 4. The values on the diagonal are the square root of AVE. The data shows that the square root of the AVE of each factor in the diagonal is greater than the corresponding correlation coefficient, thus, indicating that the scale has good discriminant validity.

**Table 4.** Matrix of square root of factor AVE values and inter-factor correlation coefficients.

|  | SN | RE | IF | PV | PR | PI |
|---|---|---|---|---|---|---|
| SN | 0.771 |  |  |  |  |  |
| RE | 0.471 | 0.799 |  |  |  |  |
| IF | 0.510 | 0.593 | 0.734 |  |  |  |
| PV | 0.468 | 0.380 | 0.374 | 0.832 |  |  |
| PR | −0.373 | −0.372 | −0.436 | −0.525 | 0.786 |  |
| PI | 0.435 | 0.483 | 0.431 | 0.504 | −0.512 | 0.777 |

Another method for discriminant validity is HTMT (heterotrait–monotrait ratio). This ratio refers to the ratio of the between-trait correlation to the within-trait correlation. It is the ratio of the mean value of the index correlation between different constructs to the mean value of the index correlation between the same constructs [52]. The evaluation method of HTMT is based on inference statistics and uses confidence intervals to measure discriminant validity, so it has its advantages. However, the Fornell–Larcker criterion is still the mainstream method.

In order to better discuss the effect of age on consumers' purchasing behavior on short-video platforms, we conduct a correlation analysis between age and the remaining variables. As shown in Table 5, we found that none of the results were significant. To investigate the reasons for this, we believe that it may be caused by the insufficient sample size and the number of research subjects in each age group is not particularly evenly distributed, thus, affecting the correlation among them.

**Table 5.** Correlation coefficient matrix between age and each variable.

|  | Age | SN | RE | IF | PV | PR | PI |
|---|---|---|---|---|---|---|---|
| Age | 1 |  |  |  |  |  |  |
| SN | −0.024 | 1 |  |  |  |  |  |
| RE | 0.072 | 0.478 ** | 1 |  |  |  |  |
| IF | −0.05 | 0.541 ** | 0.593 ** | 1 |  |  |  |
| PV | 0.013 | 0.487 ** | 0.380 ** | 0.374 ** | 1 |  |  |
| PR | −0.037 | −0.400 ** | −0.372 ** | −0.436 ** | −0.525 ** | 1 |  |
| PI | 0.078 | 0.453 ** | 0.483 ** | 0.431 ** | 0.504 ** | −0.512 ** | 1 |

Note: ** indicates $p < 0.01$.

### 4.2.2. Evaluation of Overall Model Fitness

Table 6 details the main fitness metrics obtained from the structural model test. The fit index can be used to quantify the degree of fit on the continuum. It is an overall summative statistic that evaluates how well a particular covariance structure model explains the sample data. There are many types of fit indices, such as chi-square, CFI, NNFI, RMSEA, etc., [53].

**Table 6.** Aptitude index values of the structural equation model.

| Adaptation Indicators | Recommended Value | Fitted Value |
|---|---|---|
| $\chi^2$ | The smaller the better | 254.153 |
| $\chi^2/df$ | <3.0 | 2.136 |
| CFI | >0.95 | 0.965 |
| RMSEA | <0.06 | 0.053 |
| GFI | >0.9 | 0.934 |
| AGFI | >0.9 | 0.905 |
| NNFI | >0.9 | 0.937 |
| IFI | >0.9 | 0.965 |

The Comparative Fit Index (CFI) is measured in the range of 0–1 and provides a good estimate of model fit even in small samples. A CFI value greater than 0.95 usually indicates a good model fit [50,53]. The Root Mean Square Error of Approximation (RMSEA) is the index used to evaluate the degree of misfit of the model. If it is close to 0, it means a good fit; conversely, the further away from 0, the worse the fit. An RMSEA value of 0.06 or less indicates a close fit. [54]. A value greater than 0.10 indicates a poor fit [55]. The Goodness of Fit (GFI) is the conformity between an experimental result and theoretical expectation or between data and an approximating curve. A GFI value close to 0.90 or 0.95 reflects a good fit [56]. Compared with GFI, the Adjusted Goodness of Fit (AGFI) is the value adjusted for $d_f$. AGFI value greater than 0.90 indicates a good model fit [56]. NNFI is the Non-normed Fit Index, whose value is between 0 and 1, and the closer it is to 1, the higher the degree of model fitting. When the value is greater than 0.9, the model fit degree is considered acceptable [54,57]. Incremental Fit Indices (IFI) measure how much a model has improved its fit to data. An IFI value greater than 0.9 indicates a good degree of model fitting [53,54].

By comparing with the given recommended values, the fitted values of all the fitness indicators fall within the recommended range, and therefore, the present theoretical model is acceptable.

*4.3. Results and Discussion*

4.3.1. Hypotheses Testing Results

In order to test the results of the hypothesis test, the structural relationships between the latent variables and the estimates of their standardized path coefficients are analyzed in this paper using Amos software, and the results are shown in Table 7. Table 7 shows that the path coefficients of hypotheses H2b and H3a are significant at the level of confidence $\alpha = 0.001$, hypothesis H1b is significant at the level of $\alpha = 0.01$; hypotheses H1a, H2a, and H3b are not significant.

4.3.2. Analyses of Hypotheses Testing Results

Consumers' social networks and sellers' resource endowments have a significant positive effect on consumers' purchase intention and both their local infrastructure development level has an insignificant effect on consumers' purchase intentions.

According to the research conducted by Kwun and Oh, Mehra et al. suggest that the higher awareness of brands and the guidance of opinion leaders have a positive impact on the formation of consumers' perceived values. At the same time, consumers' perceived values have a significant positive impact on their final purchase behaviors [58,59]. This is consistent with the research conclusion of this paper. In previous studies on logistics and other infrastructure, Saha et al. found that the efficiency of logistics delivery in online shopping often affects consumers' willingness to pay. The higher the distribution efficiency, the stronger the consumers' willingness to pay [16]. This differs from our findings. The reasons and mechanism for the above results are that on short-video platforms, consumers are often influenced by a variety of factors when choosing whether to make a purchase or not. Among them, the number of likes and retweets of the video and the advice of influential people around them play a big role. Compared to traditional online shopping,

short-video sales provide consumers with a more visual understanding of where fruits and vegetables are produced, and people tend to be more inclined to make purchases. However, the results of the study show that the effect of infrastructure on consumers' willingness to purchase is not significant. This may be due to the fact that the current infrastructure in China is relatively well done, and there are no major problems with express delivery and logistics, even in remote villages. Coupled with the fact that the economic development of the area we researched is, relatively, not very backward, people in that area are less likely to be bothered by such problems when choosing whether to make a purchase, leading to the failure of the previous hypothesis.

**Table 7.** Hypotheses testing results.

| Assumptions | Relationships | Standardized Path Coefficient | Conclusion |
| --- | --- | --- | --- |
| **H1a** Social network has a significant negative effect on perceived risk | SN → PR | −0.065 | Not supported |
| **H1b** Social network has a significant positive effect on consumer purchase behavior | SN → PI | 0.129 ** | Supported |
| **H2a** Resource endowment has a significant negative effect on perceived risk | RE → PR | −0.067 | Not supported |
| **H2b** Resource endowment has a significant positive effect on consumer purchasing behavior | RE → PI | 0.241 *** | Supported |
| **H3a** Infrastructure has a significant negative effect on perceived risk | IF → PR | −0.227 *** | Supported |
| **H3b** infrastructure has a significant positive impact on consumer purchase behavior | IF → PI | 0.049 | Not supported |

Note: *** indicates $p < 0.001$, ** indicates $p < 0.01$.

The level of local infrastructure development of both parties has a significant negative effect on perceived risk; consumers' social networks and sellers' resource endowments do not have a significant effect on perceived risk.

Forsythe and Shi found, in their research on consumers' perceived risks of online shopping, that the time loss caused by inconvenient infrastructure would also lead to a decrease in consumers' purchase intentions [60]. This is consistent with our research conclusion. Murray, Chen, and He found, in the study of consumers' perceived risks of shopping, that the reputation of the store and the brand awareness of the product would affect consumers' judgments of the reliability of the product, thus, reducing their perceived risk [61,62]. This differs from what we found. The reason and mechanism for the above results are that the objects of our study are fruits and vegetables, and consumers are more concerned about the logistics transportation conditions and transportation distance. They worry that the agricultural products delivered to them are not fresh enough, the quality is damaged, and the packaging is destroyed, which has a significant negative effect on the perceived risk. However, consumers' social networks and sellers' resource endowments do not hold about the negative effect on perceived risk. The reason is that the falsity of short-video marketing and some negative reports make consumers doubt the falsity of the video, even if they see the high number of likes and retweets of the video or the guarantee of the quality of the agricultural products in the live publicity, i.e., social network and resource endowment do not have a significant negative effect on the perceived risk.

### 4.3.3. Analyses of Mediating Effects

In order to examine the mediating effect of perceived value (PV) and perceived risk (PR) between social network (SN), resource endowment (RE), infrastructure (IF), and consumer purchase intention (PI), and the mediating role of perceived risk (PR) between perceived value (PV) and consumer purchase intention (PI), we used the causal stepwise regression test proposed by Baron and Kenny [63]. The inspection procedure is divided into three steps. The first step is to analyze the regression of independent variable X on dependent variable Y and test the significance of regression coefficient c. The second step is to analyze the regression of X on the mediation variable M and test the significance of regression coefficient a. The third step is to analyze the regression of X on Y, after the mediating variable M is added, and test the significance of the regression coefficients b and c′. If all the above are significant and c′ < c, it means that it is a partial intermediary. If c′ is not significant, it is a complete mediation. The results are shown in Table 8.

**Table 8.** Mediating effect test of PV and PR.

| Independent Variable | Intermediate Variables | Dependent Variable | X → Y | X + M → Y | | | Agency Situation |
|---|---|---|---|---|---|---|---|
| | | | | X → M | X → Y | M → Y | |
| SN | PV | PI | 0.453 *** | 0.487 *** | 0.273 *** | 0.371 *** | Partial mediation |
| RE | PV | PI | 0.483 *** | 0.380 *** | 0.341 *** | 0.375 *** | Partial mediation |
| IF | PV | PI | 0.431 *** | 0.374 *** | 0.282 *** | 0.399 *** | Partial mediation |
| SN | PR | PI | 0.453 *** | −0.400 *** | 0.296 *** | −0.394 *** | Partial mediation |
| RE | PR | PI | 0.483 *** | −0.372 *** | 0.340 *** | −0.386 *** | Partial mediation |
| IF | PR | PI | 0.431 *** | −0.436 *** | 0.256 *** | −0.400 *** | Partial mediation |
| PV | PR | PI | 0.504 *** | −0.525 *** | 0.325 *** | −0.342 *** | Partial mediation |

Note: *** indicates $p < 0.001$.

Bootstrapping is another way to test for mediating effects, which can be applied to small- and medium-sized samples. This method refers to whether the 95% confidence interval of the product term (a*b) of regression coefficient a and regression coefficient b includes the number 0; if the 95% confidence interval does not include the number 0, it indicates a mediating effect. If the 95% confidence interval includes the number 0, there is no mediating effect [64]. Compared to the previous method, the method of bootstrapping is more complex and difficult to understand, so it is not as widely used as the causal stepwise regression method.

The test results show that after adding the mediating variable perceived value, the positive influence of social network on consumer purchase intention decreases; the positive influence of resource endowment on consumer purchase intention decreases; and the positive influence of infrastructure on consumer purchase intention decreases. Thus, perceived value plays a partially mediating role between social network and consumers' willingness to buy, resource endowment and consumers' willingness to buy, and infrastructure and consumers' willingness to buy. This is consistent with the research conclusion of Parasuraman et al. that we mentioned above [39]. Partial mediation means that social network, resource endowment, and infrastructure can influence consumers' buying behavior of agricultural products directly, or by influencing consumers' perceived value and, thus, their final buying behavior. It follows that hypothesis H4 is verified.

With the inclusion of the mediating variable perceived risk, the positive effect of social network on consumer purchase intentions decreases; the positive effect of resource

endowment on consumer purchase intentions decreases; the positive effect of infrastructure on consumer purchase intentions decreases; and the positive effect of perceived value on consumer purchase intentions decreases. Thus, perceived risk partially mediates between social network and consumer purchase intention, between resource endowment and consumer purchase intention, between infrastructure and consumer purchase intention, and between perceived value and consumer purchase intention, i.e., hypothesis H5 is tested. This is consistent with the research conclusion of Agheshlouei et al. that we mentioned above [40].

## 5. Conclusions

### 5.1. Research Findings

Based on 406 valid data obtained from online and offline channels, we analyze the purchasing behavior of consumers on short-video platforms, explore the influence of social network, resource endowment, and infrastructure on their purchasing behavior of agricultural products using the SEM model, and examine the mediating role played by perceived value and perceived risk, with the help of Amos software. We find that two factors, social network and resource endowment, significantly influence consumers' purchasing behaviors toward agricultural products; the positive influence of good resource endowment is the most significant, followed by social network, and the influence of infrastructure is relatively insignificant. The negative effect of infrastructure on consumers' perceived risk is more significant. In addition, the results of the mediating effect test indicate that perceived value and perceived risk have a mediating effect between external stimulus variables and consumers' purchasing behavior of agricultural products, i.e., social network, resource endowment, and infrastructure can further influence consumers' purchasing intention through consumers' perceptions.

### 5.2. Policy Recommendations

Based on the above findings, we propose the following policy recommendations to help farmers make better use of short videos for marketing, so as to improve their income level and better respond to China's policy of comprehensively promoting rural revitalization, accelerating the modernization of agriculture and rural areas, and comprehensively developing the agriculture, the countryside, and farmers. In addition, by analyzing the factors influencing consumers' purchase of agricultural products on short-video platforms, we hope to increase consumers' purchase behavior by improving the quality of short-video and strengthening supervision by merchants and platforms, thus, promoting the healthy development of the short-video industry and providing a theoretical basis for the implementation of industrial poverty alleviation and digital countryside strategy, as well as the development of "Internet+" agriculture.

#### 5.2.1. For Short-video Merchants

First, we find that consumers' social networks have a positive impact on their perceived values and purchase intentions. Therefore, short-video merchants need to strengthen their publicity and expand the influence of their own accounts with a view to increasing the number of likes, attention, and retweets of their stores so that more potential consumers can see them and increase the likelihood of purchase. In order to expand their influence, merchants need to improve their own reputation and accumulate store popularity. The team needs to have professional staff to learn how to effectively operate the store on the platform, and then gradually promote the short videos and live broadcasts, etc., released by the store, and encourage consumers to recommend more to their friends around them, forming a virtuous cycle of mutual promotion.

Secondly, the resource endowment and infrastructure of short-video merchants can significantly affect the purchasing behavior of consumers. So, merchants should choose suitable anchors and live locations when they broadcast. For example, fruits and vegetables can be anchored by local experienced farmers, and the location of live broadcast can be

diversified, even live picking can be done in the farmland. Stores can show more of the production and planting place of the agricultural products they sell during the live broadcast, so that consumers can understand the production source of the agricultural products in a more realistic way. In addition, when making short videos, merchants can consider matching the corresponding text to attract consumers' attention, and at the same time, merchants can also consider making the agricultural products for sale in the form of short films with episodes to show consumers the origin of the agricultural products.

### 5.2.2. For Short-Video Platforms

From the research results, we know that perceived risk has a significant negative impact on consumers' purchase intentions. Therefore, the short-video platform should strengthen the management of the platform merchants, develop reward and punishment mechanisms, punish the merchants with a poor reputation, and give some rewards to the merchants with a good reputation, so as to force the merchants to improve their own reputation, avoid false marketing, and ensure that the real products match the pictures.

The perceived value of consumers has a significant positive impact on their willingness to buy, so it is necessary to promote the merchants to improve their own live standards to improve the perceived value of consumers. For this reason, the short-video platform should monitor and evaluate the live broadcast activities of the merchants. For example, the short-video platform can score the live broadcast behavior and effect of merchants, put forward corresponding suggestions for each live broadcast, and push excellent live broadcast shops worth learning for them.

**Author Contributions:** Conceptualization, S.J. and Z.W.; methodology, S.J. and Z.W.; software S.J.; validation, J.R. and Z.S.; formal analysis, S.J. and Z.S.; investigation, S.J. and Z.W., Z.S. and J.R.; data curation, J.R. and Z.S.; writing—original draft preparation, S.J.; writing—review and editing, Z.S. and J.R.; funding acquisition, Z.S. and J.R. All authors have read and agreed to the published version of the manuscript.

**Funding:** This research was funded by the National Key Research and Development Program of China (2020YFD1100601), Natural Science Basic Research Plan in Shaanxi Province of China under Grant (2022JM-419), Special Fund of Basic Scientific Research Fund of Central Universities under Grant (Research on Agricultural Economy and Management Innovation under the Background of Rural Revitalization, 2452022061) and National Natural Science Foundation of China under Grant (71973106); in part by the Shaanxi Science Fund for Distinguished Young Scholars under Grant (2021JC-21); in part by the 2021 Key Scientific Research Project of Shaanxi Education Department under Grant (21JT043); in part by the Special Fund of Basic Scientific Research Fund of Central Universities under Grant (2452022070).

**Institutional Review Board Statement:** Not applicable.

**Informed Consent Statement:** Not applicable.

**Data Availability Statement:** The data presented in this study are available on request from the corresponding author.

**Conflicts of Interest:** The authors declare no conflict of interest.

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
