# Peer review of "Determinants of Buying Produce on Short-Video Platforms: The Impact of Social Network and Resource Endowment—Evidence from China"

_agriculture, doi:10.3390/agriculture12101700_

Round 1

Reviewer 1 Report (Previous Reviewer 3)

Thank you for the opportunity to review your paper again. I feel the paper is much improved. Here I am referring in particular to the introduction, literature/ conceptual framework including hypothesis development, and the conclusion section. However, what is not yet completely sorted is the method section- even though I like the presentation of the questionaire design, the explanation related to external stimilus variables and the partial mediation.

1) Can you please outline elaborate on your approach to structural equation modeling? Can you elaborate on measurement and structural model and the analysis?

2) Which school /approaches/ authorgroups are you following? Can please indicate this through proper citation? 

2) Aligned with my previous comment, can you please also indicate the proper citations for threshhold values. That readers can better follow up and put in perspective what you have done here.

4) Can you comment on bootstrapping, Fornell–larcker criterion and heterotrait multitrait ratio, fit indeces? 

Otherwise, the work has been genuinely improved and I have no major reservation towards the paper. Looking forward to see the improved manuscript

Author Response

Reviewer 2 Report (New Reviewer)

The study elucidates the impact of social network and resource endowment factors on the determinants of buying produce via short video platforms. It has very important implication for business in decision making both for buyers and sellers, but authors need to address the following points.

Title:

Title is very important and should be very clear statement for readers’ understanding. In this study used determinants and factors both in the title, it is suggested that delete the word “factors”, please revise it.

Abstract:

Need to revise and improve grammar of English language. For instance, line 9, short video selling…. , it should be selling by short video.

Line no. 10-15, sentence is too long, authors need to divide into two sentences.

Introduction:

Overall, connection of the background is good but English language need to improve.

Such as line No.30~35, 2022 is written many times in one sentence, further sentence is too long, this sentence may divide into two sentence. Line No.66 the reference No.5, I suggest to keep the reference at the end of the sentence but keep scholars name as the same place. The same suggestion for reference 7, 8 and other which are cited likewise. Lines No.76~80, the sentence is too long and used few words many times in a sentence, so it needs to revise. Line 80~84 can be divided into two sentences. Lines No. 104~110, too long sentence, need to divide into two or more sentences.

Authors contribution No.3, Lines No. 111~123, explained the results of present study? If yes, I think no need to explain your study results in this section, that would be in the results section, conclusion and abstract section. But authors can explain the contribution in different words not as results here. For instance, this research contributes to analyze the social network and resource endowment on perceived risk and infrastructure effect on consumer’s willingness to purchase.

Literature Review:

Line 204, Chen and Wang wanted to study….., the word “wanted” not appreciate to use, authors should use alternate word, such as Chen and Wang conducted study on willingness to purchase…..

There are some other grammar errors, need to revise.

Conceptual framework:

This section is appropriate and written well.

Materials and Methods:

Lines 367~370 have English grammar mistakes, need to revise. Such as “after the questionnaire was designed” is not correct.     

Results and Discussion:         

Discussion is weak, authors need to cite more references to compare their study results with the existing literature.

Conclusion:

Conclusion is fine

References:

List of references is ok but as I mentioned above, authors need to cite more references in the result section for debate to compare with existing literature.

Round 2

Reviewer 1 Report (Previous Reviewer 3)

All my comments have been carefully adressed. I have no more reservation towards the paper. Well done!

This manuscript is a resubmission of an earlier submission. The following is a list of the peer review reports and author responses from that submission.

Round 1

Reviewer 1 Report

Dear Authors, 

Thank you for in interesting paper. After going over the paper, I have some minor recommendations regarding the content of the paper, that I feel could help improve it: 

- on line 25, you mention JD and PDD, I recommend presenting the name fully and after using an abreviation;

- on lines 37 - 42 , you mention market evolution in Yuan, I would also add the value in US Dollars, in order for international readers to understand the market size; 

- Lines 45 - 58 represent the same paragraph as the previous one

- Line 104 - You mention "most scholars`` without mentioning who they are, please reference

- line 146, you mention the 4R marketing - I recommend that you present and explain what the 4R's are. 

- In the Data collection component you mention offline random sampling, could you explain the method of random sampling. Also you mention online and offline data collection methods, could you please mention the percentage of responses from the online and from the offline

 - For the Descriptive statistics - did you also had the `Age` component in your research? If so could you also include that in your findings. It would be interesting to see the age of the respondents correlated with the results. 

- In my opinion the 5.1 section could be presented in a more extensive way. And it would be interesting if the conclusions also underline some potential theoretical implications not only business policy recommendations. 

Reviewer 2 Report

Under the COVID-19 pandemic, citizens were forced to live in isolation and transform their behavioral lifestyle. As a result, the number of viewers of short-form videos has increased, and citizens' information fighting behavior has also been transformed. In this context, short-form video sales, the focus of this study, is expected to continue to develop as a new business model that enhances mutual communication between buyers and sellers.

The quantitative analysis method (structural equation model) applied by this study in the framework of stimulus-organization-response (SOR) is convincing, and the proposed model and measures provide useful tools for further research in this field.

In addition, the adoption and demonstration of the consumer's social network, the seller's resource endowment, and the level of infrastructure development on both sides as "key components of external stimuli" in the analytical model speaks to the contribution that these model frameworks can make in presenting valid results and implications.

At the same time, the study's focus on the mediating role played by consumers' perceived value and perceived risk as one of the study's unique features is commendable, as it is well grounded in the study's theme and the characteristics of the short-form video business.

As a general comment, however, it is necessary to critically evaluate the finding of the obtained analysis that "enhanced monitoring and management of platforms is the key to stable growth of consumers' willingness to purchase agricultural products" in light of existing literature studies, and to present the implications of this study more clearly. In addition, the logic that leads to the stable growth of the said business and the increase in the income level of rural households based on this finding is somewhat too subjective. We would like to see a more objective and persuasive editorial based on empirical results.

<Some more comments and suggesitons>

Table 6: Hypothesis testing results>>>Please use the term 'supported' not support.

4.2.2. Research based on structural equation modeling>>>What are the dot numbers, are they sub-headings?  

Table 7: Mediating effect test of PV and PR>>>plesse explain what 'agents' mean.

5.2. Policy Recommendations>>>here again, please double check the presentaiton style. For an academic paper, you do not need to add section number (not a sub-heading?) for each paragraph. Please follow any other published articles in Agriculture to keep coherent and readable style.

Good luck!

Reviewer 3 Report

Dear Authors, I read your study and I have the following comments: 

1. Please go through the manuscript and cited appropriately. In large parts that is missing.

2. Your approach to structural equation modeling is not at all explained. Who do you follow Hair ? Readers not familiar will not be able understand if the approach and proceeding is explained. This will put results in perspectivem

Round 2

Reviewer 3 Report

Some general information about structural equation modelling was included. But again, which approach do you follow? And why you have not cited on the new paragrahs included?